# High Mobility Group Box-1 and Diabetes Mellitus Complications: State of the Art and Future Perspectives

**DOI:** 10.3390/ijms20246258

**Published:** 2019-12-11

**Authors:** Federico Biscetti, Maria Margherita Rando, Elisabetta Nardella, Andrea Leonardo Cecchini, Giovanni Pecorini, Raffaele Landolfi, Andrea Flex

**Affiliations:** 1U.O.C. Clinica Medica e Malattie Vascolari, Fondazione Policlinico Universitario A. Gemelli IRCCS, 00168 Roma, Italy; g.pecorini@gmail.com (G.P.); raffaele.landolfi@unicatt.it (R.L.); andrea.flex@unicatt.it (A.F.); 2Laboratory of Vascular Biology and Genetics, Università Cattolica del Sacro Cuore, 00168 Roma, Italy; 3Università Cattolica del Sacro Cuore, 00168 Roma, Italy; m.margheritarando@gmail.com (M.M.R.); elisabetta.nardella@gmail.com (E.N.); alcech92@gmail.com (A.L.C.)

**Keywords:** High Mobility Group Box-1 (HMGB1), diabetes mellitus, vascular complications

## Abstract

Diabetes mellitus (DM) is an endemic disease, with growing health and social costs. The complications of diabetes can affect potentially all parts of the human body, from the heart to the kidneys, peripheral and central nervous system, and the vascular bed. Although many mechanisms have been studied, not all players responsible for these complications have been defined yet. High Mobility Group Box-1 (HMGB1) is a non-histone nuclear protein that has been implicated in many pathological processes, from sepsis to ischemia. The purpose of this review is to take stock of all the most recent data available on the role of HMGB1 in the complications of DM.

## 1. Introduction

Diabetes mellitus (DM) is a chronic, metabolic disease representing globally the ninth major cause of death. It involves the 10% of world’s population and it is characterized by hyperglycemia due to a defective insulin secretion or insulin resistance [1]. Chronic inflammation plays a key role in the pathogenesis of DM and several recent studies have analyzed the relation between the High Mobility Group Box-1 (HMGB1) protein and DM, demonstrating its pivotal role on the disease progression. The aim of this review is to summarize the current knowledge about HMGB1 and its linkage with DM complications.

## 2. Diabetes Mellitus

DM is a chronic disease and its prevalence is increasing worldwide, representing a major public health problem. According to the World Health Organization (WHO), DM affected 422 million adults aged over 18 years in 2014, representing the seventh leading cause of death in 2018 [2]. A poor control of the disease leads to development of cardiovascular complications and to an increased risk of premature death, with a relevant impact on healthcare and a high economic burden [2]. Most DM sufferers are affected by type 2 diabetes (T2DM), the most widespread form of DM, characterized by hyperglycemia due to insulin resistance and pancreatic beta-cell dysfunction [3]. Several studies focused their attention on the role of inflammation in the pathogenesis of DM. In particular, many authors demonstrated that elevated levels of C-reactive protein (CRP), IL-6, TNF-α predict the development of T2DM [4,5,6,7,8,9]. Hotamisligil and colleagues found that levels of TNF- α are elevated in the adipose tissue of obese insulin-resistant rodents and obese humans, and that the neutralization of TNF-α in insulin-resistant rodents resulted in an increase peripheral uptake of glucose in response to insulin [10,11]. The role of TNF-α in insulin resistance seems to be related to a reduced expression of the insulin-sensitive glucose transporter GLUT4. In fact, TNF-α promotes the reduction of insulin receptor substrate 1 (IRS-1) mRNA and GLUT4mRNA, leading to insulin resistance and hyperglycemia [12] Moreover, Massaro and coworkers showed that peroxisome proliferator activated receptor (PPAR) alpha/gamma agonists attenuated insulin resistance in human adipocytes, reducing pro-inflammatory mediators including IL-6, CXC-L10 and monocyte chemoattractant protein (MCP-1), supporting the pathogenic role of inflammation in DM development [13]. Hyperglycemic environment is even characterized by enhanced production of reactive oxygen species (ROS), formation of advanced glycation end products (AGEs), activation of protein C kinase (PCK), and activation of polyol pathway [14]. All these factors promote a pro-inflammatory cytokines milieu, including TNF-α, IL-1β, IL-6- IL-8 and HMGB1, which contribute to endothelial damage, development of atherosclerosis and impaired angiogenesis, leading actors in diabetic vascular complications [15].

## 3. HMGB1 and Diabetes

HMGB1 is a DNA-binding protein that belongs to the High mobility group (HMG) superfamily, a group of ubiquitous non-histone nuclear proteins, identified for the first time in 1973 by Goodwin and Johns and characterized by high mobility in polyacrylamide gel electrophoresis [16]. HMG can be divided in three groups: HMGB, HMGN and HMGA [17,18]. HMGB family comprises HMGB1, HMGB2, HMGB3 and SP100HMG [15,19,20,21] and it is characterized by the HMG box, a particular DNA-binding motif that defines this particular group of nuclear proteins [20]. In particular, HMGB1 is a 30 kDA nuclear protein composed by 215 amino acids containing two N-terminal DNA-binding domains, called BOX A and BOX B, and an acidic C-terminal tail [22,23,24]; BOX B is, in general, responsible of the pro-inflammatory effect stimulating the release of cytokines [25]. Conversely, BOX A seems to attenuate the inflammatory cascade [15]. Inside the cell nucleus, HMGB1 has both a structural role and a role in DNA transcription, replication and repair; it also contributes to nuclear proteins assembly [26]. In the cytoplasm, it acts as a signaling regulator and, in the extracellular milieu, it is involved in inflammatory cascade, acting as an “alarmin” and as a pro-inflammatory cytokine [26]. Moreover, HMGB1 contributes to cell migration and proliferation, cell differentiation and tissue regeneration [3,20,25], taking part in different pathophysiological processes and diseases, such as sepsis, arthritis, cancer, atherosclerosis, diabetes and cardiovascular diseases [19,27,28,29,30,31]. HMGB1 is translocated outside the cell in case of cellular damage or cellular death and it was also clearly shown that it can be actively secreted by stimulated immune cells such as monocytes, macrophages, mature dendritic (MD) cells, natural killer (NK) cells and endothelial cells as a result of different stimuli, such as exposure to lipopolysaccharide (LPS), TNF-α, or IL-1β, IFN-γ and tissue injury [3,19,25,32,33,34]. Furthermore, it has been demonstrated that oxidative stress influences the release of HMGB1 [35]. Interestingly, Lu and colleagues showed that translocation from nucleus to cytoplasm, is mediated by JAK/STAT1 pathway [36], and translocation from cytoplasm to extracellular milieu depends on a mechanism mediated by inflammasome activation during pyroptosis, a form of pro-inflammatory programmed cell death [36]. In diabetes condition, hyperglycemia promotes directly or indirectly, through ROS and AGEs production, the release of HMGB1 [32]. It has been demonstrated that HMGB1 is overexpressed in diabetic patients, compared to non-diabetic [37,38,39]. Moreover, Dandona and coworkers showed that, in patients affected by type 1 DM (T1DM), insulin with glucose infusion suppressed ROS generation, toll-like receptors (TLRs), HMGB1 and other signaling molecules, while glucose infusion alone increased ROS generation, expression of HMGB1 and plasma concentrations of HMGB1 [40]. Once in extracellular space, pro-inflammatory effect of HMGB1 is mediated by its extracellular receptors, in particular receptor for advanced glycation end products (RAGE) and toll-like receptor (TLR)-2, -4, and -9. Recently, other receptors have been studied, such as C-X-C chemokine receptor type 4 (CXCR4), macrophage antigen-1, syndecan-3, CD24, Siglec-10, T cell Igmucin-3 [41], even if their role in the HMGB1 signaling pathway is not completely clear.

### 3.1. The Role of RAGE

RAGE is the first-discovered HMGB1 receptor and it acts as receptor of other ligands, such as AGEs, β-amyloids, S100 proteins, C3a-anaphylatoxin and LPS [3,15]. It is expressed by different cells, including monocytes, macrophages, fibroblasts, smooth muscle cells, endothelial cells and cancers cells, and it plays a pivotal role in diabetes pathogenesis and progression [3]. Binding RAGE, HMGB1 activates CDC42/Rac and mitogen-activated protein kinase (MAPKs) pathways, in particular p38 mitogen-activated protein kinase (p38MAPK), and extracellular signal-regulated kinase (ERK), leading to nuclear translocation of activated nuclear factor-kB (NF-kB) [32]. NF-kB promotes the release of different cytokines (including TNF-α, IL-6, IL-1), growth factors and adhesive molecules responsible for endothelial cells activation, chemotaxis and maturation of immune cells [3,25,32]. RAGE is also involved in ROS formation, enhancing oxidative stress by NADPH oxidase activation [42].

### 3.2. The Role of TLRs

TLRs belong to the family of type I transmembrane receptors, composed of an extracellular ligand-binding domain, a single transmembrane domain and a cytosolic Toll-interleukin 1 receptor (TIR) domain [43]. They are expressed in endothelial cells, smooth cells, macrophages, hepatocytes, airway epithelial cells, adipose tissue and skeletal muscle and they are involved in innate immune response, recognizing pathogen associated molecular patterns (PAMPs) and damage associated molecular patterns (DAMPs) [15,43]. HMGB1 interacts in particular with TLR-2, TLR-4 and TLR-9, promoting the activation of NF-kB, IL-1 associated kinases (IRAK) and IkB kinases (IKK), via the myeloid differentiation factor-88 (MyD88) pathway [15]. Moreover, TLR-4 and TLR-3 initiate interferon regulatory factor 3 (IRF3) and NFkB with consequent activation of interferon-inducible genes [15]. TLRs also act via p38 MAPK and ERK 1 and 2, leading to activation of NF-kB [3]. The interaction between HMGB1 and its receptors (particularly, RAGE and TLRs) enhances both inflammatory response and immune response, elements that promote cellular damage and vascular dysfunction, leading to development of atherosclerosis and complications of DM. Interestingly, HMGB1 is also involved in DM initiation, altering insulin secretion and insulin resistance. In particular, interaction with RAGE promotes pancreatic islet cell apoptosis in a way dependent by enhanced oxidative stress [32]. Moreover, the binding with TLR-4 enhances insulin resistance via the phosphorylation of peripheral insulin receptor substrate [32]. In addition, Zhang and colleagues showed that HMGB1 was passively released from late apoptotic pancreatic beta-cells, leading to an autoimmune response against beta-cells self-antigens and it is involved in islet allograft rejection of diabetic patients after islet transplantation [44]. Furthermore, it has been shown that HMGB1 plays a role in adipose tissue, activating macrophages and stimulating release of IL-6 and TNF-α. Therefore, HMGB1 contributes also to metabolic dysfunction linked to insulin resistance [3]. As mentioned above, through the linkage to its receptors, HMGB1 promotes ROS formation and it activates the pro-inflammatory cascade responsible of endothelial damage and progression of atherosclerosis. HMGB1 has also a pro-angiogenic role [32]. In fact, its secretion by activated macrophages enhances release of angiogenic factor, as vascular endothelial growth factor (VEGF), TNF-α and IL-8 [45], which promote mobilization of endothelial progenitor cells (EPCs), precursors of endothelial cells involved in angiogenesis and endothelial repair [32,45]. Furthermore, a study by Wu and coworkers, demonstrated that oxidative stress induced by AGEs, promoted release of HMGB1 from EPCs. The same study showed that inhibition of HMGB1 was related to decreased oxidative stress induced by AGEs, suggesting a role of HMGB1 in amplifying ROS production [46]. Interestingly, redox environment seems to affect also HMGB1 action in extracellular milieu [35]. In fact, depending on the redox state, HMGB1 can be present in three different forms based on the oxidative state of cysteine residues of the molecule: all-thiol HMGB1, disulfide HMGB1 and fully oxidized HMGB1 [35]. All-thiol HMGB1 induces autophagic response through RAGE, moreover it has chemotactic properties binding CXCR4. Disulfide HMGB1 induces innate immune response binding TLR-4 and, conversely, fully oxidized HMGB1 has not a pro-inflammatory function, inducing immune tolerance [35,47,48]. In addition, HMGB1 is able to promote vascular calcification, enhancing osteochondrogenic differentiation of cells such as vascular smooth muscle cells (VSMCs), pericytes, myofibroblasts, progenitor cells [41], via TLR4-JNK-NFkB pathway [49]. Moreover, it promotes via RAGE signaling the expression of TGF-β that is associated with fibrosis and calcification [50,51].

In the following sections, the available evidence on the role of HMGB1 and diabetic complications will be discussed.

## 4. Diabetic Macroangiopathy

Macrovascular diabetic complications, in particular coronary artery disease (CAD), cerebrovascular disease (CVD) and peripheral artery disease (PAD), affect 20–30% of patients with T2DM [52]. Compared to non-diabetic, diabetic patients have a more aggressive atherosclerosis and an impaired mechanism of vascular repair [53]. Summarized data indicate that the most important mechanisms linked to diabetic macrovascular injury are activation of polyol and hexosamine pathways and AGEs production [54]. In particular, AGEs, binding RAGEs, promote activation of different pathways, above all NF-kB signaling, resulting in increased oxidative stress and release of cytokines, molecules of adhesion and pro-angiogenic factors from endothelial cells, macrophages and VSMCs. Hyperglycemia and ROS production also promote TLR-2 and TLR-4 activation, with release of TNF-α, IL-1β, IL-8, monocyte chemoattractant protein (MCP-1), vascular cell adhesion molecule 1 (VCAM-1), intracellular adhesion molecule 1 (ICAM-1). Moreover, a cooperative interaction between RAGE and TLRs could enhance inflammation and immune response in diabetic milieu [55]. Furthermore, as mentioned above, hyperglycemia acts directly on HMGB1 release which enhances ROS production, endothelial cells dysfunction and VSMCs damage [32,40]; HMGB1 acts in synergy with TLRs and RAGE, promoting progression of atherosclerosis and DM vascular complications. Even the abnormal release of nitric oxide (NO), which characterized the diabetic environment, directly promotes endothelial dysfunction [56]. In addition, hyperglycemia promotes calcification in atherosclerotic plaque and in the tunica media of large and medium size arteries, through a major mobilization of osteoprogenitor cells, from bone marrow to the vascular wall [41,54,57,58,59]. Moreover, hyperglycemia can promote vascular injury with an indirect action of other risk factors, such as low-density lipoprotein (LDL) cholesterol [54]. Diabetic milieu contributes, in fact, to oxidative modification of LDL (Ox-LDL). Ox-LDL promote themselves ROS production, by NADPH oxidase activation, and expression of pro-inflammatory cytokines and adhesion molecules, enhancing atherosclerosis plaque formation [60]. Diabetic patients have also a dysfunction of stem/progenitor cells involved in vascular homeostasis [54]. Indeed, hyperglycemia and insulin resistance affect progenitor cells function, resulting in impaired mobilization of stem cells (“bone marrow mobilopathy”) [54] from bone marrow to peripheral vessels, leading to a defective regeneration of endothelial dying cells and VSMCs after injury [54]. In diabetic macroangiopathy, there is also an impaired neovascularization and intra-plaque angiogenesis, with an increased vascular permeability of capillary vessels. These factors contribute to plaque instability, promoting hemorrhage and plaque rupture [54,61]. Several evidences have analyzed the role of HMGB1 in plaque formation, rupture and thrombosis, which represents the underlying pathogenic mechanisms of macrovascular complications of DM.

## 5. HMGB1 and Diabetic Coronary Artery Disease

Diabetes mellitus represents a major risk factor for cardiovascular diseases [62]. In particular, T2DM-related cardiovascular diseases are the main causes of mortality and morbidity worldwide [63]. As mentioned above, activation of inflammatory cells, with increased release of cytokines and chemokines, plays a pivotal role in atherosclerosis progression [37]. Different studies demonstrated that HMGB1 levels were correlated with non-calcific atherosclerotic coronary plaque [64] and severity of CAD [65]. Benlier and coworkers showed that HMGB1 levels were higher in patients with CAD, compared with healthy individuals, suggesting that HMGB1 was an independent risk factor for the disease; however, in this study 62% of patients with CAD were affected by T2DM [66]. Moreover, with development of atherosclerosis from fatty streaks to fibro-fatty lesion, the number HMGB1-producing macrophages considerably increase in atherosclerotic plaque, demonstrating a role of HMGB1 in disease progression [67]. In addition, HMGB1 seems to indirectly promote CAD, in particular enhancing the pro-inflammatory effect of epicardial adipose tissue on coronaries wall by RAGE signaling [68]. Also, patients with ST-elevation myocardial infarction have higher serum HMGB1 levels that are associated with entity of the ischemic injury, pump failure and increased mortality [69,70,71]. Conversely, in rats with experimental myocardial infarction, inhibition of HMGB1 promoted thinning and expansion of the infarct scar and marked hypertrophy of the non-infarcted area, maybe through impairment of the infarct-healing process depending on HMGB1 function [69,72]. Another interesting article showed that mice with overexpression of HMGB1, after ligation of left anterior descending coronary artery, had better survival rates, a smaller size of myocardial infarction, and cardiac remodeling and dysfunction were prevented, compared to control mice. Moreover, HMGB1 overexpression improved capillary and arteriole formation after ischemic injury [73]. HMGB1 is also involved in ischemia-reperfusion injury of the heart, influencing the entity of damage. In fact, inhibition of HMGB1 reduces infarct size and markers of tissue damage, while treatment with recombinant HMGB1 worsens ischemic-reperfusion injury [74]. According to the evidences already discussed, Yan and coworkers demonstrated a linkage between CAD and HMGB1 serum levels in diabetic and non-diabetic patients, showing that HMGB1 levels was elevated in diabetic and non-diabetic patients with CAD [37]. Furthermore, HMGB1 levels correlated with IL-6, TNF-α and high sensitivity C-reactive protein (hsCRP) and patients with CAD had a decreased level of esRAGE [37], which could correlate with an enhanced inflammatory state [75].

In particular, the hsCRP activity on p38MAPK pathway induced the expression of HMGB1, which promoted release of other cytokines and it enhanced the atherosclerotic process, supporting a chronic inflammation of vessels wall [37]. Conversely, Yin and colleagues showed that HMGB1 levels were higher in patients with CAD and T2DM compared with non-diabetic patients with CAD, and that HMGB1 expression correlated with levels of Hemoglobin A1c [76]. Hemoglobin A1c values was also correlated with the severity of CAD [77], so HMGB1 could represent a marker of CAD degree in T2DM patients. Another study demonstrated that CD34-positive EPCs, cells with an anti-thrombotic action by facilitating thrombi organization, were reduced in fragments of aspirated thrombi of diabetic patients with acute myocardial infarction compared to non-diabetic ones [78]. Indeed, HMGB1 levels were localized in the leukocyte nuclei and extracellular area of all thrombi and, after immunofluorescence staining, extracellular HMGB1 immunopositive area was larger in diabetic patients, compared with non-diabetic. These data suggest a role of HMGB1 in facilitating thrombus formation, in particular in diabetic patients, worsening the prognosis of patients with acute myocardial infarction [78] (Table 1). Therefore, HMGB1 affects progression and severity of CAD and it could be involved in major adverse cardiac events (MACE) in diabetic patients after acute myocardial infarction; however, further studies are needed to verify this hypothesis.

## 6. HMGB1, Diabetes Mellitus and Peripheral Arterial Disease

Peripheral arterial disease (PAD), and in particular the lower extremities (LE) PAD is a form of atherosclerosis that affects the arteries of the lower limbs, leading to several complications such as critical limb ischemia, acute limb ischemia, tissue necrosis and gangrene [79,80]. It represents a major public health problem and, in 2010, 202 million people globally was affected by PAD [81], with a prevalence of 20–30% in patients with DM [82], the major risk factor for the disease [14]. Different mechanisms contribute to the development of PAD in a diabetic environment and inflammatory process has been widely recognized as the major player of atherosclerotic progression of lower limbs [57,79,83,84]. In fact, the pro-inflammatory milieu, typical of DM, with increased levels of cytokines, chemokines, adhesion molecules and proteolytic enzymes [60], together with enhanced production of ROS and AGEs formation, is responsible for endothelial and vascular damages at the base of PAD progression. Moreover, it has been recently demonstrated that elevated levels of osteoprotegerin, TNF-α, IL-6 and CRP correlate with worse vascular outcomes, after revascularization, in diabetic patients with PAD [83]. Regarding HMGB1, it has been shown that patients with PAD undergoing LE surgery have increased levels of HMGB1 in muscle tissue, compared to patients without PAD [85]. Moreover, during experimental acute limb ischemia/reperfusion in rats, HMGB1 promoted vascular structure remodeling and vasomotor dysfunction, through activation of inflammatory cascade, degradation of collagenous and elastic fibers [86]. Hyperglycemic environment affects levels of HMGB1, which is responsible of atherosclerosis progression. In fact, serum levels of this nuclear protein are higher in diabetic patients with PAD, compared to diabetic patients without PAD, and plasma levels of HMGB1 correlated with clinical severity of the disease [57,87]. Also, it has been demonstrated that patients with diabetic foot ulceration have higher serum levels of cytokines such as TNF-α, IL-6 and HMGB1, compared to diabetic patients without foot ulceration [88]. Despite these evidences, HMGB1 seems to be essential also in tissue repair after peripheral ischemia [89]. In fact, in diabetic mice with experimental model of limb ischemia, HMGB1 levels are decreased in ischemic tissues; furthermore, administration of HMGB1 improves neovascularization via a VEGF-dependent mechanism [45]. Moreover, non-diabetic mice treated with anti-HMGB1 antibodies have a poor recovery after muscle ischemia [85]. Thus, HMGB1 represents a molecule involved both in vessels injury and in repair process, favoring post-ischemic angiogenesis. Decreased levels of this protein could affect peripheral outcomes after revascularization. Interestingly, data by Xu and coworkers demonstrated that in mice with experimental limb ischemia, administration of chloroquine improves perfusion recovery and it induces HMGB1 release in sarcoplasm of muscle tissue and in serum [85]. It has been evaluated even the role of HMGB1 in cell therapy with human adipose-derived stem cells (HASCs) that are able to differentiate into EPCs with release of angiogenic and growth factors, given their stromal vascular fraction [89]. In particular, inhibition of HMGB1 in mice treated with human adipose-derived stem cells (HASCs) reduces VEGF levels and blood flow recovery, suggesting that HMGB1 positively influences HASCs treatment, promoting post-ischemic angiogenesis [89]. However, in relation to the results of discussed evidences (Table 2), further research is needed to clarify the role of HMGB1 in PAD, and other clinical studies are needed to evaluate the role of this protein in patients with PAD subjected to revascularization of the lower limb in terms of outcome.

## 7. HMGB1 and Diabetic Cerebrovascular Disease

Stroke represents an important cause of disability and mortality worldwide. Diabetic patients have a higher risk in incurring in a cerebrovascular accident and hyperglycemia is correlated to worse outcome after stroke [90]. Commonly to other macrovascular complications of DM, pathogenesis of diabetic CVD is linked to different mechanisms, such as vascular endothelial dysfunction, increased arterial stiffness and systemic inflammation. These factors are related both to atherosclerosis of cerebral vessels and to a worse prognosis after an acute event [90,91]. In fact, after an ischemic injury, activation of the inflammatory response promotes in the early phase brain tissue damage and brain blood barrier leakage, worsening post-stroke outcome [92]. Conversely, in the later phase inflammation contributes to repair of neuronal cells [93]. In particular, patients with ischemic stroke show higher expression of TLR-2 and TLR-4 that correlates with higher serum levels of IL-1β, IL-6, TNF-α, VCAM-1 and that is associated with infarct volume and poor outcome, enhancing brain inflammatory injury [94]. In addition, HMGB1 is involved in ischemic stroke with a role in cerebrovascular injury. Specifically, Qiu and coworkers showed that in mice with experimental middle cerebral artery occlusion, in the early stage of ischemic injury, HMGB1 was released from nucleus and cytoplasm of neurons into extracellular space, promoting expression of TNF-α, NO synthetase and ICAM-1 in neurons, astroglia and endothelial cells, by binding RAGE and TLRs [93]. HMGB1 is also involved in blood-brain-barrier permeability impairment, worsening outcome after ischemic injury [92].

Based on the evidences discussed above, it is conceivable that in the later phase of stroke a different oxidative environment could contribute to the role of inflammation in repair of neuronal cells [93] acting on HMGB1 [35,47,48].

Today, there are still few evidences available on the role of HMGB1 in diabetic CVD (Table 3). However, like for other diabetic complications, it is possible to hypothesize a key role of the molecule on diabetic stroke [92,95]. In streptozotocin-induced diabetic rats, middle cerebral artery occlusion resulted in increased levels of serum HMGB1 and Matrix Metallopeptidase-9 (MMP-9) and increased expression of matrix metallopeptidase-9 (MMP-9), RAGE and TLR-4 in brain tissue, in particular in microglia. Moreover, treatment with niaspan, a slow-release form of niacin, reduced levels of HMGB1, MMP-9, RAGE, TLR-4 [95]. Niacin is related to a protective effect after cerebrovascular injury, reducing inflammation and improving endothelial function. It increases angiopoietin-1 levels that belongs to a group of growth factors involved in neuronal differentiation, vascular remodeling and endothelial cells survival [95,96]. Hence, niaspan could represent a treatment option to improve neurological outcome after stroke. Interestingly, a work by Hu and coworkers demonstrated that intravenous injection of bone marrow stromal cells into T2DM rats with experimental middle artery occlusion reduced HMGB1 and RAGE expression in ischemic brain, moreover it improved functional recovery and it attenuated blood brain barrier leakage by increasing levels of desmin and ZO-1 [92]. These data are apparently in contrast with a previous study conducted on T1DM rats with middle artery occlusion. In fact, Chen and coworkers showed that bone marrow stromal cells treatment decreased survival of diabetic rats with experimental stroke, it increased blood–brain barrier leakage, brain hemorrhage and vascular density. Furthermore, it promotes atherosclerosis increasing intimae thickness and collagen production in internal carotid artery [97]. Due to the conflicting results of the two studies, further works are needed to elucidate the role of bone marrow stromal cells in diabetic stroke, being a possible therapeutic strategy in treatment of diabetic CVD.

## 8. Diabetic Microangiopathy

In addition to macrovascular complications of DM, mechanisms involved in diabetic microangiopathy are generally represented by ROS production, activation of polyol, PKC-diacylglycerol and hexosamine pathways, AGEs formation and activation of inflammatory process [98]. In particular, the main feature of diabetic microangiopathy is the increased permeability of vessels and an altered angiogenesis mechanism, due to an impaired function of endothelial cells, VSMCs, stromal cells, pericytes, vascular stem cells and inflammatory cells [98]. Given the interest matured towards HMGB1 and its relationship with DM pathogenesis, several studies have evaluated its role in development and progression of diabetic nephropathy, retinopathy and neuropathy.

## 9. HMGB1 and Diabetic Nephropathy

Diabetic nephropathy (DN) represents the leading cause of end-stage kidney disease in the world and it affects 30–40% of people with DM [99]. Clinically, DN is characterized by proteinuria, hypertension, and reduction in kidney function [99] which are the outcome of kidney injury induced by hyperglycemia. Diabetic kidney damage is firstly characterized by renal hypertrophy due to accumulation of extracellular matrix proteins in glomerulus, with consequent glomerular basement membrane thickening and mesangial matrix expansion for accumulation of laminin, collagen, fibronectin via TGF-β. There is also a reduction of nephrin levels, a receptor protein essential for the function of the glomerulus [100]. All these factors lead to glomerulosclerosis and tubulointerstitial fibrosis, features of the end stage disease [101]. As for other diabetic complications, kidney injury is strictly related to inflammation, and different studies have shown that plasma levels of HMGB1 are increased DN, suggesting a role of the protein in DN pathogenesis (Table 4). Specifically, Kim and coworkers described the linkage between HMGB1, RAGE and NF-kB pathway, demonstrating that HMGB1 was highly expressed in renal glomerular cells and tubular epithelial cells of diabetic rats, in particular in the nucleus and cytoplasm of cells, compared with normal controls, where HMGB1 was expressed only in the nucleus [102]. Diabetic rats even showed elevated levels of RAGE and NF-kB. These factors were involved in kidney damage through regulation of different mediators, as TNF-α, IL-6, IL-1, ICAM-1, Granulocyte-Macrophage Colony-Stimulating Factor (GM-CSF) [102]. Other studies have shown that DN is characterized by higher levels of TLRs and HMGB1 and other cytokines such as TNF-α, IL-6, IL-1β, TGF-β1, ICAM-1, MCP-1 [103,104,105,106,107,108]. In particular, Lin and coworkers demonstrated that in renal tubules of subjects with diabetic nephropathy, there is increased expression of TLR-4 and HMGB1, but not of TLR-2 [103]. They showed also that levels of TLR-4 correlated with interstitial macrophage infiltration and glycosylated hemoglobin. Moreover, in vitro hyperglycemia induced TLR-4 expression via PKC leading to increased levels of IL-6 and chemokine (CC motif) chemokine ligand 2 (CCL-2), via IkB/NF-kB pathway [103]. Conversely, knockdown of TLR gene in mice induced reduction of IL-6 and CCL-2 levels, leading to a protective effect on kidney [103]. These evidences support the key role of HMGB1 and its link with TLRs network in nephropathy pathogenesis. In contrast with the previous work, Mudalier and coworkers showed that high glucose exposure increased even levels of TLR-2 and HMGB1 [104]. Moreover, HMGB1 stimulated the NF-kB pathway. In fact, silencing of TLR-2 interrupted NF-kB nuclear expression and HMGB1-induced NF-kB-DNA binding [104]. Interestingly, a study by Zhang and coworkers demonstrated that glycyrrhizic acid (GA), an HMGB1 inhibitor, reduced expression of HMGB1, RAGE, TLR-4 and activation of ERK, p38 MAPK and NF-kB pathways in kidney tissue of diabetic rats and it also reduced levels serum and kidney levels of TNF-α, IL-6, IL-1β, MCP-1, ICAM-1, TGF-β1 [108]. Moreover, a recent study by Jigheh and coworkers showed that empaglifozin reduced renal levels of HMGB1, RAGE, TLR-4 alleviating renal inflammation, suggesting a possible therapeutic approach to the disease [109].

## 10. HMGB1 and Diabetic Neuropathy

Diabetic neuropathy (DNr) is a degenerative disease and it represents globally the most widespread form of neuropathy [110]. DNr involves different kind of nerves, including large and small sensory fibers, autonomic and motor nerves, and it is responsible for several debilitating complications such as pain, foot ulcers, gangrene, amputations, gastrointestinal, genitourinary and cardiovascular dysfunctions [110,111]. It occurs in different forms and, depending on the anatomical distribution, we can distinguish two subgroups of this disorder: diffuse neuropathies (principally, diabetic polyneuropathy and diabetic autonomic neuropathy) and focal neuropathies (diabetic cranial neuropathy, diabetic mononeuropathy, radiculoplexus neuropathies) [111]. Regarding DNr pathogenesis, several mechanisms are implicated in damage of myelinated structures and neurovascular injury, in particular formation of ROS, activation of polyol pathway and formation of AGEs, release of pro-inflammatory cytokines (IL-1β, IL-6, TNF-α), activation of poly(ADP ribose) polymerase (PARP) and activation of hexosamine pathway and PKC pathway [110]. There are also implicated down-regulation of growth factors, up-regulation of neurotrophin 3, autophagic pathway and wnt pathway [110]. Few studies have analyzed the linkage between HMGB1 and DNr [112,113,114,115,116] (Table 5), starting from previous data that demonstrated the role of HMGB1 in nervous system [117,118,119]. In fact, as mentioned above, HMGB1 is involved in central ischemic damage, being released into the extracellular space after ischemic insult where it promotes neuroinflammation [117]. Inhibition of HMGB1 expression also reduces infarct size and microglia activation [117]. Moreover, in rat models, HMGB1 promotes pain hypersensitivity after peripheral nerve injury, probably through RAGE activation. Conversely, anti-HMGB1 antibody treatment alleviates hyperalgesia. Nerve injury also increases expression of HMGB1 mRNA in dorsal root ganglia (DRG) and spinal nerve [118]. In addition, Zhao and coworkers showed that calmodulin-dependent protein kinase IV (CaMKIV), a protein kinase involved in neuropathic pain, and HMGB1 were upregulated in DRG of rats treated with streptozotocin (STZ), used to induce diabetes and neuropathic pain. Moreover, inhibition of phosphorylated CaMKIV (pCaMKIV) decreased HMGB1 levels and reduced thermal hyperalgesia and mechanical allodynia in diabetic rats, confirming, however, the role of HMGB1 in neuropathic pain [114]. The expression of HMGB1 in neuropathic pain seems also correlated with Sigma-1 receptor (Sigma-1R), a receptor involved in nociception. In fact, STZ treatment induces Sigma-1R and HMGB1 expression in DRG, with increased tactile allodynia and thermal hyperalgesia in rat models [112]. Conversely, knockdown rats for Sigma-1R show a modest tactile allodynia and thermal hyperalgesia in absence of increased cytoplasmic levels of HMGB1 levels [112]. The same result has been demonstrated after HMGB1 inhibition in spinal cord. These findings suggest a role of Sigma-1R in promoting HMGB1 expression [112].

The relationship between HMGB1 and neuropathic changes induced by hyperglycemia has been evaluated even in retinal neuropathy [113,116]. In fact, DM induces upregulation of HMGB1, activation of ERK1/2 pathway, activation of Cleaved Caspase-3, an apoptosis executer enzyme, and glutamate signaling pathways in rat’s retinas. In diabetic retinas, there are even decreased levels of Glyoxalase-1 (GLO1), an important enzyme for AGEs detoxification. Moreover, GA intake decreased HMGB1 mRNA levels in diabetic rats compared to controls rats [116]. HMGB1 affects even cells survival, decreasing retinal ganglion cells survival [113]. However, we need further studies to confirm the role of HMGB1 in DNr and to provide a valid therapeutic option to stop progression of DNr.

## 11. HMGB1 and Diabetic Retinopathy

Diabetic retinopathy (DR) represents the leading cause of blindness globally and it is characterized by retinal microvascular injury and neuronal damage, even if the temporal relationship between vascular and neuronal changes remains unclear [120,121]. Microvascular injury of DR is well characterized and consists of a non-proliferative phase with occlusion of retinal vessels leading to ischemic retinal areas and macular edema, and in a proliferative phase characterized by increased vasopermeability and proliferation of retinal vessels and hemorrhages [120]. Different mechanisms are implicated in the pathogenesis of DR, specifically, generation of ROS and AGEs, activation of polyol and hexosamine pathways, enhanced cells apoptosis and release of cytokines (IL-1β, TNF-α), adhesion molecules (ICAM-1), leukocyte infiltration, abnormal expression of growth factors and activation of nuclear factors pathways [122,123]. All these factors promote alteration of retinal neurons and glial cells with impaired control of glutamate metabolism, impaired synaptic activity, neuronal apoptosis and activation of microglia with enhancement of inflammatory response [120]. As a mediator of inflammatory and angiogenic processes, several studies have analyzed the role of HMGB1 in DR pathogenesis (Table 6).

Levels of HMGB1 are increased in retinas of diabetic rats [123,124] and in patients with proliferative diabetic retinopathy (PDR) [125,126], together with endothelial activity biomarkers like ICAM-1, sICAM-1, MCP-1, VEGF, Granulocyte-colony stimulating factor (G -CSF), sVE-cadherin and soluble endoglin (sEng), supporting the contribution of HMGB1 in neovascularization process [123,125,126,127,128,129]. Moreover, El-Asrar and coworkers demonstrated that levels of HMGB1 are higher in patients with PDR and retinal hemorrhages, suggesting a role of the protein in PDR progression [126]. The pro-inflammatory action of HMGB1 even in retinal cells is promoted by the linkage to TLR-4 [130,131] and RAGE [122,131,132,133], and the activation of ERK1/2 and NF-kB signaling [132]. In fact, the intravitreal administration of the protein enhances these pathways and downregulates TLR-2 and occludin expression, enhancing retinal vasopermeability [134]. Furthermore, the link of HMGB1 to RAGE and TLR-4 seems to inhibit insulin signaling in retinal cells [131]. Despite this evidence, a study by Chang and colleagues showed that, in human retinal pigment epithelial cells, HMGB1 upregulated expression of VEGF, basic fibroblast growth factor (bFGF), transforming growth factor β2 (TGF-β2), connective tissue growth factor (CTGF) and phosphorylation of Akt, p38MAPK and NFkB, but it did not mediate ERK, cJun N-terminal kinase (JNK) and Smad2 (130). Interestingly, two studies suggest that HMGB1 has an indirect role in retina and choroidal neovascularization [127,128]. In particular, they showed that HMGB1 mediates endothelial cells directly, while pericytes death, responsible of vasopermeability and endothelial proliferation, was mediated by HMGB1 cytotoxic activity on glial cells, through TLR-4-dependent production of ROS and cytokines [127,128]. Moreover, subretinal injection of HMGB1 in rats did not induce neovascularization and it did not modify expression of VEGF-A in glial cells [128]. Furthermore, it has been shown that HMGB1 promotes oxidative stress in retina. In particular, Mohammad and coworkers showed that HMGB1 induced NADPH-oxygen-derived ROS, causing retinal cells apoptosis [134]. They also showed that HMGB1 induced cleaved caspase-3, IL-1β and PARP-1, and that this effect is inhibited by administration of GA. Interestingly, in a work by Sohn and coworkers, the administration in diabetic rats of extract of Polygonum cuspidatum (PCE), a dried root with anti-inflammatory action, reduced HMGB1, RAGE and NF-kB expression and it ameliorated vascular retinal permeability, inhibiting tight junction leakage [133]. To support evidences of HMGB1 activity on progression of DR, Jiang and coworkers demonstrated that intravitreal injection of HMGB1 siRNA in rats reduced retinal damage and cellular death and it improved retinal function [135]. Furthermore, in human retinal endothelial cells treated with high glucose, HMGB1siRNA reduced oxidative stress and cellular apoptosis. In addition, a recent study demonstrated that exosomes derived from mesenchymal stem cells overexpressing miRNA-126, an endothelial specific miRNA that mediates inflammation, suppressed HMGB1 expression and NF-kB and NLRP3 inflammasome activity in human retinal endothelial cells [136,137]. Finally, protein kinase A (PKA) seems to inhibit cytoplasmic HMGB1 [137].

## 12. HMGB1 and Diabetic Cardiomyopathy

Diabetic cardiomyopathy (DC) is a form of myocardial dysfunction leading to heart failure and it occurs in diabetic patients, independently from coronary artery disease and hypertension [138]. Hyperglycemia plays a key role in the development of this disorder, inducing oxidative stress and release of several cytokines such as IL-1β, TNF-α, TGF-β1, which promote myocardial cell death and fibrosis. However, pathogenesis of DC is not completely understood [139]. In vitro and in vivo, hyperglycemia induces HMGB1 RNA expression and it increases HMGB1 protein levels in myocardial cells and fibroblasts [139,140]. Furthermore, in diabetic mice with post-myocardial infarction remodeling, there are increased levels of HMGB1 with enhanced inflammation and fibrosis and it has been demonstrated that knockdown of HMGB1 and RAGE genes reduces inflammation and infarct size [140]. Moreover, mice knockdown for RAGE showed a decreased expression of HMGB1 and a reduced activity of NF-kB in heart, demonstrating that the action of HMGB1 is mainly mediated by RAGE and NF-kB in myocardial cells too [140]. In diabetic mice, HMGB1 is also responsible for cardiomyocytes apoptosis through activation of ERK/Ets-1 pathway, involved in cell growth, proliferation and cellular apoptosis. In particular, inhibition of HMGB1 with HMGB1siRNA reduces ERK and Ets-1 phosphorylation induced by hyperglycemia [141]. In addition, HMGB1 is involved in myocardial fibrosis [139,142]. In particular, Wang and coworkers showed that HMGB1 increased TGF-β1 levels in cardiac fibroblasts and it enhanced MMP activity and collagen I and collagen III expression. They also demonstrated that inhibition of HMGB1 reduced p38MAPK, ERK1/2 and JNK signaling, which are crucial pathways in cardiac hypertrophy, fibrosis and cytokine-mediated inflammation [141]. Interestingly, Song and coworkers demonstrated that cardiac expression of HMGB1 was mediated by PI3Kgamma/Akt pathway in a hyperglycemic environment and treatment with an antioxidant prevented PI3Kgamma/Akt signaling and HMGB1 production. This finding provides a new pathway that mediates HMGB1 expression, representing a possible therapeutic target for treatment of DC [143]. Furthermore, a recent study by Wu and coworkers showed that in diabetic mice treatment with resveratrol induced a cardioprotective effect, reducing HMGB1 expression and downregulating RAGE, TLR-4, NF-kB signaling. Resveratrol also reduced oxidative stress and ameliorated myocardial fibrosis and inflammation, reducing TNF-α and iNOS levels. These findings support further data on resveratrol and its cardioprotective effect, by reducing oxidative stress [144], and representing a possible strategy of treatment for cardiac dysfunction (Table 7, Figure 1).

## 13. Conclusions

DM represents an endemic disease with an important impact on healthcare and a high social cost. Several convincing data show that HMGB1 plays a pivotal role in the pathogenesis of all diabetic complications. Few works have also analyzed the effect of different molecules in attenuated HMGB1 signaling (GA, resveratrol, niaspan), suggesting possible new therapeutic strategies. Because of the increased morbidity and mortality related to diabetic complications and thanks to the identification of new molecular mechanisms underlying pathogenesis of DM, it is therefore essential to focus further research on novel therapeutic approaches for the treatment of DM and prevention of its complications.

## Figures and Tables

**Figure 1 ijms-20-06258-f001:**
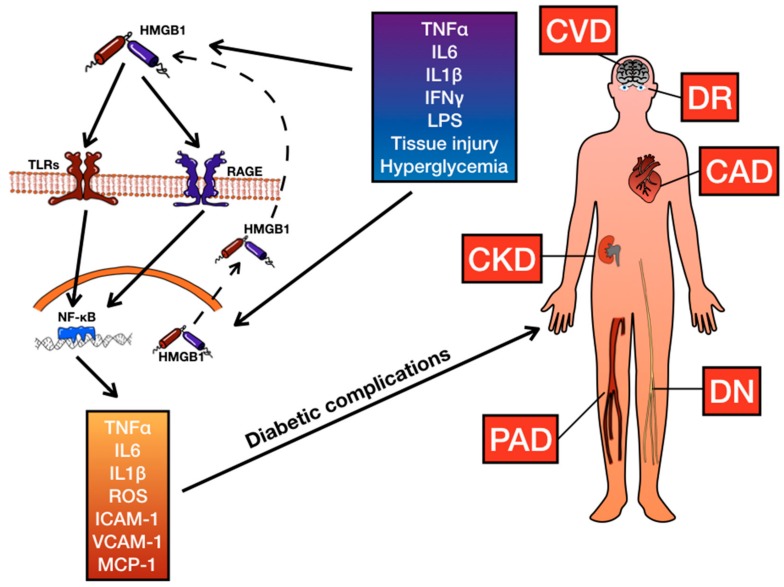
Effect of High mobility group box-1 (HMGB1) on inflammation and vascular complications of diabetes. HMGB1 system influences risk of coronary artery disease (CAD), cerebrovascular disease (CVD), diabetic retinopathy (DR), chronic kidney disease (CKD), diabetic neuropathy (DN), and peripheral artery disease (PAD). TLRs, Toll-Like Receptors; RAGE, Receptor for Advanced Glycation End Products; IL1β, interleukin 1β; IL6, interleukin 6; TNFα, Tumor Necrosis Factor-α; NFkB, nuclear factor kappa-light-chain-enhancer of activated B cells; ROS, Reactive oxygen species; ICAM-1, Intercellular Adhesion Molecule 1; VCAM-1, Vascular Cell Adhesion Molecule 1; MCP-1, Monocyte Chemoattractant Protein 1; IFNγ, Interferon γ; LPS, Lipopolysaccharide.

**Table 1 ijms-20-06258-t001:** High Mobility Group Box-1 (HMGB1) and diabetic coronary artery disease, summary of the evidences. CAD: coronary artery disease; T2DM: type 2 diabetes mellitus; hsCRP: high sensitivity C-reactive protein; TNF-α: Tumor Necrosis Factor-α; IL-6: interleukin 6; HbA1c: glycosylated Hemoglobin A1c.

Ref.	Year	Aim	HMGB1
Yan et al.	2009	Role of HMGB1 in T2DM patients with CAD e in non-T2DM patients with CAD.	Increased serum levels of HMGB1 and hsCRP in non-T2DM patients with CAD and in T2DM patients with CAD. HMGB1 levels correlate with hsCRP, TNF-α, IL-6 levels.
Yamashita et al.	2012	Presence of CD34-positive cells and HMGB1 in acute coronary thrombi in T2DM patients vs non-T2DM patients.	Extracellular HMGB1 area in the thrombi larger in T2DM patients. HMGB1 facilitates thrombus formation.
Yin et al.	2015	Relationship between blood glucose levels and HMGB1 levels T2DM patients with CAD.	Increased serum levels of HMGB1 in T2DM patients with CAD. Levels of HMGB1 correlate with glucose levels and HbA1c levels.

**Table 2 ijms-20-06258-t002:** HMGB1, diabetes mellitus and peripheral arterial disease, summary of the evidences. STZ: streptozotocin; VEGF: vascular endothelial growth factor; T2DM: type 2 diabetes mellitus; PAD: peripheral arterial disease; OPG: osteoprotegerin; TNF-α: Tumor Necrosis Factor-α; IL-6: interleukin 6; CRP: C-reactive protein; HASCs: human adipose-derived stem cells; DFU: diabetic foot ulceration.

Ref.	Year	Aim	HMGB1
Biscetti et al.	2010	Role of HMGB1 in diabetic angiogenesis.	Lower HMGB1 protein expression in the ischemic tissue of STZ-induced mice. HMGB1 administration ameliorates the blood flow recovery and capillary density in the ischemic muscle of STZ-induced mice. Reduced HMGB1-induced angiogenesis by inhibiting VEGF activity.
Tsao et al.	2015	Expression of HMGB1 in diabetic foot atherogenesis.	Increased HMGB1 expression in vessels of T2DM patients and T2DM patients with PAD compared to non-T2DM patients.
Giovannini et al.	2017	Role of HMGB1, OPG, TNF-α, IL-6, CRP in diabetic patients with PAD.	Increased serum levels of HMGB1 in T2DM patients with PAD. Levels of HMGB1 positively correlates with severity of PAD in T2DM patients. HMGB1 independent risk factor for PAD in T2DM patients.
Biscetti et al.	2017	Role of HMGB1 in cell therapy with HASCs in PAD.	Improved blood flow recovery in mice co-treated with HASCs and HMGB1 protein, compared to HASCs-treated mice. Reduced post-ischemic angiogenesis with HMGB1 inhibition in mice co-treated with HASCs and HMGB1.
Hafez et al.	2018	Role of Sirtuin-1 and HMGB1 in DFU.	Increased serum levels HMGB1 and AGEs in T2DM patients compared to non-T2DM patients with the highest levels in T2DM patients with DFU.

**Table 3 ijms-20-06258-t003:** HMGB1 and diabetic cerebrovascular disease, summary of the evidences. T1DM: type 1 diabetes mellitus; MCAo: middle cerebral artery occlusion; T2DM: type 2 diabetes mellitus; BMSCs: bone marrow stromal cells; N: nicotinamide; BBB: blood brain barrier.

Ref.	Year	Aim	HMGB1
Ye et al.	2011	Role of HMGB1 and RAGE in stroke in T1DM rats and role of niaspan on pro-inflammatory proteins expression.	Increased HMGB1 expression after stroke in brains of STZ-induced MCAo rats. Niaspan treatment in STZ-induced MCAo rats decreased HMGB1 expression.
Hu et al.	2016	Role of HMGB1 in stroke in T2DM rats and role of BMSCs in HMGB1 inflammation.	MCAo STZ/N-induced rats: increased HMGB1 and RAGE expression, increased BBB leakage, decreased functional outcome after stroke. Injection of BMSCs in STZ/N-induced rats: decreased HMGB1 and RAGE expression, attenuated BBB leakage and improved functional outcome after stroke, decreased inflammation after stroke.

**Table 4 ijms-20-06258-t004:** HMGB1 and diabetic nephropathy, summary of the evidences. Nuclear factor kappa-light-chain-enhancer of activated B cells: NF-kB; DN: diabetic nephropathy; STZ: streptozotocin; TLR: Toll-Like Receptor; T2DM: type 2 diabetes mellitus; WT: wild type; GA: glycyrrhizic acid.

Ref.	Year	Aim	HMGB1
Kim et al.	2011	Role of HMGB1, RAGE and NF-kB in DN.	Increased HMGB1 levels in cytoplasmic and nuclear patterns of glomerular cells of STZ-induced rats vs non STZ-induced rats (nuclear HMGB1 only).
Lin et al.	2012	Role of TLR-4 in DN.	Increased HMGB1 expression in renal biopsies of T2DM patients with DN.
Mudaliar et al.	2013	Role of TLR-2 and -4 in DN.	Upregulation of HMGB1 and TLR-2 levels in tubules cells of STZ-induced mice compared to non STZ induced mice. Increased HMGB1 secretion and NF-kB activation in response to high level of glucose. Reduced HMGB1 secretion and NF-kB activation with TLR-2 siRNA and TLR-4 siRNA.
Ma et al.	2014	Role of TLR-4 in DN.	Upregulation of HMGB1 in STZ-induced WT mice and STZ-induced TLR-4 deficient mice compared to non STZ-induced mice.
Ma et al.	2014	Role of TLR-2 in DN.	Upregulation of HMGB1 in STZ-induced WT mice compared to non STZ-induced WT mice.
Chen et al.	2015	Role of high glucose on HMGB1 expression in T2DM patients and in mesangial cells.	Increased serum HMGB1 levels in T2DM patients and in mesangial cells stimulated with high glucose. Knockdown of HMGB1 in mesangial cells reduces HMGB1 mRNA levels.
Zhang et al.	2017	Role of HMGB1 inhibitor GA in DN.	Increased HMGB1 expression in kidney tissue of STZ-induced rats and lower HMGB1 expression in kidney tissue of STZ-induced rats treated with GA.
Jigheh et al.	2018	Empaglifozin role in reduction of HMGB1 and TLR-4 levels in DN.	Empaglifozin reduces renal levels of HMGB1 in STZ-induced rats.

**Table 5 ijms-20-06258-t005:** HMGB1 and diabetic neuropathy, summary of the evidences. STZ: streptozotocin, GA: glycyrrhizic acid; TLR: Toll Like Receptor; GWAS: genome-wide association study; NP: neuropathic pain; RGC: retinal ganglion cells; CaMKIV: calmodulin-dependent protein kinase IV; DRG: dorsal root ganglia; TA: tactile allodynia, TH: thermal hyperanalgesia.

Ref.	Year	Aim	HMGB1
El Asrar et al.	2014	Role of HMGB1 in retinal neuropathy.	Increased HMGB1 retinal levels in STZ-induced rats. Vitreal injection of HMGB1in non STZ-induced rats increased HMGB1 mRNA levels. GA intake in STZ-induced rats decreased HMGB1 mRNA levels.
Meng et al.	2015	GWAS to clarify the role of sex specific involvement of Chr1p35.1 (ZSCAN-TLR12P) and Chr8p23.1 (HMGB1P46) in diabetic NP.	Involvement of Chr8p23.1 (HMGB1P46) in NP, with high heritability of Chr8p23.1 (HMGB1P46) in males.
Zhao et al.	2015	Role of HMGB1 in RGC in high glucose environment.	Increased levels of HMGB1 mRNA and protein in RGC and decreased cells survival in high glucose environment. Decreased levels of HMGB1 mRNA and protein in RGC and increased cells survival after siRNA HMGB1 injection.
Zhao et al.	2016	Role of CaMKIV in diabetic NP and its relationship with HMGB1 expression in DRG.	Increased pCAMKIV and HMGB1 levels in DRG of STZ-induced rats. Inhibition of CAMKIV reduced CAMKIV and HMGB1 expression in DRG of STZ-induced rats.
Wang et al.	2018	Role of sigma 1 receptor and its relationship with HMGB1 expression in DRG in NP.	In STZ-induced rats TA and TH correlate with increased HMGB1 expression in DRG. Stimulation of Sigma-1R induces TA and TH and increases HMGB1 levels. Blockade of Sigma-1R and HMGB1 reduce TA and TH. STZ-induced Sigma-1R knockdown rats have modest NP and no variation in HMGB1 levels.

**Table 6 ijms-20-06258-t006:** HMGB1 and diabetic retinopathy, summary of the evidences. PDR: proliferative diabetic retinopathy, MCP-: ICAM-1: intracellular adhesion molecule-1; sICAM-1: soluble intracellular adhesion molecule-1; IL-1β: interleukin 1β; GM-CSF: Granulocyte-Macrophage Colony-Stimulating Factor; G-CSF: Granulocyte-colony stimulating factor; sVE-cadherin: soluble vascular endothelial-cadherin; sEng: soluble endoglin; DR: diabetic retinopathy, GA: glycyrrhizic acid; ERK1/2: extracellular signal–regulated kinase1/2; NFkB: nuclear factor kB; nuclear factor kappa-light-chain-enhancer of activated B cells; PLA-2: phospholipases A2; VEGF: vascular endothelial growth factor; HMERC: human retinal microvascular endothelial cells; Nox: NADPH oxidase; bFGF: basic fibroblast growth factor; TGF-β2: Transforming growth factor β2; MAPK: Mitogen-Activated Protein kinase; PARP-1: poly (ADP-ribose) polymerase-1; TLR: Toll Like Receptor; RGC: retinal ganglion cells; PCE: Polygonum cuspidatum, JNK: c-Jun N-terminal kinase; HRECs: human retinal endothelial cells; PKA: protein kinase A; MSC-exos: mesenchymal stem cells-derived exosomes.

Ref.	Year	Aim	HMGB1
El-Asrar et al.	2011	Levels of HMGB1, RAGE in patients with PDR and correlation with MCP-1, sICAM-1, IL-1 beta, GM-CSF.	Increased HGMB1 levels in vitreous samples of patients with PDR. Increased HMGB1 levels in patients with PDR and hemorrhages.
El-Asrar et al.	2012	Levels of HMGB1 in patients with PDR and correlation with VEGF, G-CSF, sVE-cadherin, sEng.	Increased HMGB1 levels in vitreous samples of patients with PDR.
Mohammad et al.	2012	Role of HMGB1 in DR.	Increased HMGB1 levels in retinas of STZ-induced rats. Intravitreal administration of HMGB1 in non STZ-induced rats increases levels of ICAM-1, sICAM-1, HMGB1, RAGE, ERK1/2, NFkB and retinal permeability. Administration of GA reduced upregulation of HMGB1 in STZ-induced rats.
Gong et al.	2014	Role of HMGB1 and PLA2 in DR.	Increased HMGB1 levels in retinal tissue of STZ-induced rats; HMGB1 induces endothelial cells death directly and pericytes death through cytotoxic activity of glial cells.
Santos et al.	2014	Role of HMGB1 in vulnerability of endothelial cells and pericytes.	HMGB1 induces endothelial cells death directly and pericytes death through glial cells. No differences between neovascularization and levels of VEGF after HMGB1 subretinal administration. No involvement of HMGB1 in rats with oxygen induced retinopathy.
Fu et al.	2015	Serum HMGB1 and VEGF levels in DR patients.	Increased serum HMGB1 levels in DR patients. In vitro HMGB1 inhibits human retina pigment epithelium cells growth and it induces apoptosis.
Mohammad et al.	2015	Relationship between HMGB1 and NADPH oxidase-derived ROS in DR.	Increased HMGB1 levels and oxidative stress in vitreous fluid of PDR patients. HMGB1 enhances IL-1β, ROS, Nox2, PARP-1, and cleaved caspase-3 production by HRMEC. Diabetes and intra-vitreal injection of HMGB1 in normal rats induce significant upregulation of ROS, Nox2, PARP-1, and cleaved caspase-3 in the retina.
Yu et al.	2015	Role of HMGB1 in DR inflammation and cellular apoptosis.	Increased HMGB1 expression in retinas of STZ-induced rats. HMGB1 accelerates apoptosis of diabetic retinal cells.
Kim et al.	2016	Role of HMGB1, RAGE, NFkB in DR.	Increased HMGB1 cytoplasmic translocation in high glucose environment.
Jiang et al.	2016	Role of HMGB1 and TLR-9 in DR.	Increased HMGB1 and TLR-9 expression in retinas tissues and in RGC of STZ-induced rats.
Sohn et al.	2016	Protective effect of PCE in DR by inhibition of HMGB1 pathway.	Increased levels of HMGB1 in retinas of STZ-induced rats. Treatment with PCE reduces HMGB1 and RAGE expression in retinas of STZ-rats.
Chang et al.	2017	Role of hypoxia in HMGB1 release in DR.	Hypoxia induces HMGB1 cytoplasmic release. HMGB1 upregulates expression of VEGF, bFGF, TGF-β2, CTCF and phosphorylation of Akt, p38MAPK and NFkB but not ERK, JNK, Smad2. HMGB1 causes growth suppression and G1 cell cycle arrest in ARPE-19 cells. Neutralization of TLR4 and RAGE reduces HMGB1-driven cytokines production.
Jiang et al.	2017	Role of HMGB1siRNA in DR.	HMGB1siRNA reduces apoptosis and oxidative damage of retinal cells in STZ-induced rats (intravitreal injection) and in HRECs (HMGBsiRNA pre-treatment) of treated with high. Decreased IKKβ and NFκB protein expression after HMGB1 silencing.
Jiang et al.	2018	Role of HMGB1on the proteins involved in insulin signaling.	Recombinant HMGB1 blocks insulin receptor and Akt phosphorylation through RAGE and TLR-4.
Liu et al.	2018	Role of PKA in HMGB1 inhibition.	PKA inhibits cytoplasmic HMGB1, activating IGFBP-3 and SIRT1.
Zhang et al.	2018	Role of exosome derived from mesenchymal stem cells in retinal inflammation reduction.	MSC-exos overexpressing miR-126 suppresses HMGB1 expression and NLRP3 inflammasome activity in human retinal endothelial cells.

**Table 7 ijms-20-06258-t007:** HMGB1 and diabetic cardiomyopathy, summary of the evidences. STZ: streptozotocin; Ets-1 E26 transformation-specific sequence-1; ERK1/2: extracellular signal–regulated kinase1/2; TGF-β1: Transforming growth factor β1; MMP: matrix metallo-proteinase; MAPK: mitogen-activated protein kinase; IL-33: interleukin 33; DC: diabetic cardiomyopathy; PI3Kγ: phosphatidylinositol 3-kinase-gamma.

Ref.	Year	Aim	HMGB1
Volz et al.	2010	Role of HMGB1 in diabetic heart disease.	High glucose treatment of cardiac fibroblasts, macrophages and cardiomyocytes increased HMGB1 mRNA expression and protein levels. Increased cardiac HMGB1 mRNA expression and protein levels in STZ-induced mice with post-myocardial infarction remodeling and HMGB1 blockage reduces post-myocardial infarction remodeling and markers of tissue damage.
Wang et al.	2014	Role of HMGB1 in high glucose-induced apoptosis of cardiomyocytes.	HMGB1siRNA reduces cell apoptosis in high glucose milieu through Ets-1/ERK1/2 signaling.
Wang et al.	2014	Role of HMGB1 in fibrosis and myocardial dysfunction.	HMGB1 silencing ameliorated LV dysfunction and remodeling in STZ-induced mice. High glucose milieu induced HMGB1 translocation and secretion in isolated cardiac fibroblasts. Administration of HMGB1 increased expression of collagens I and III and TGF-β1 in cardiac fibroblasts. HMGB1 inhibition reduced high glucose-induced collagen production, MMP activity, proliferation, MAPK signaling.
Tao et al.	2015	Extracellular communication pathways between cardiomyocytes and fibroblasts in DC.	Increased myocardial expression of HMGB1, collagen deposition and myocardial dysfunction and reduced IL-33 in STZ-induced mice. Inhibition of HMGB1 prevents myocardial collagen deposition and dysfunction. Increased HMGB1 secretion and collagen I production in high glucose induced cardiomyocytes/fibroblasts. HMGB1 inhibition reduces collagen I expression in the fibroblasts.
Song et al.	2016	Intracellular signaling pathway leading to cardiomyocyte HMGB1 expression in hyperglycemia.	Increased HMGB1 expression in high glucose-conditioned cardiomyocytes by PI3Kγ and Akt pathway. Treatment of cardiomyocytes with an antioxidant abolished high glucose-induced PI3Kγ and Akt activation and HMGB1 production.
Wu et al.	2016	Expression of HMGB1 pathway and oxidative stress in resveratrol-treated diabetic mice.	Lower serum and bone marrow-derived monocytes HMGB1 levels in STZ-induced rats treated with resveratrol.

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
