# Peer review of "High Mobility Group Box-1 and Diabetes Mellitus Complications: State of the Art and Future Perspectives"

_ijms, 2019, doi:10.3390/ijms20246258_

Round 1

Reviewer 1 Report

This is a clear, well-written paper that gives a brief summary of most recent data on the role of HMGB1 protein in diabetic complications. The similar review has been reported previously  (*e.g. Wu H, Chen Z, Xie J, Kang L-N, Wang L, Xu B. High Mobility Group Box-1: A Missing Link between  Diabetes and Its Complications. Mediators of Inflammation. 2016;2016:3896147). However, the paper of Biscetti and colleagues provides additional, novel information about the involvement of HMGB1 in diabetes related pathologies, such as in coronary artery disease and foot ulceration, which were published after 2016. Thus, this paper can be attracting for a researchers, especially those interested for new therapeutic approaches.

Minor point:

1.      Page2, line 80: inflammasome instead inflammosome.

2.      Please be consistent and use either abbreviation or spelled out words (Page 11, line 409:    GA /glycyrrhizic acid)

Author Response

Authors’ response

We thank the Reviewer for his/her encouraging comments.

Minor point:

Page2, line 80: inflammasome instead inflammosome.

Authors’ response

The wrong word has been corrected throughout the manuscript.

Please be consistent and use either abbreviation or spelled out words (Page 11, line 409:    GA /glycyrrhizic acid)

Authors’ response

The abbreviation has been preferred throughout the manuscript.

Reviewer 2 Report

Major comments

1. Authors describe that inflammatory stimuli enhance HMGB-1 expression or release in diabetes. However, inflammation is common pathophysiology in atherosclerosis and its complications. Please describe possible mechanisms of diabetes specific HMGB-1 expression and/or cell damage/death in macrovascular complications.

2. Authors simply summarize effects of HMGB-1 on diabetic complications in Figure 1. Macrovascular, microvascular, and diabetic cardiomyopathy show distinct pathology. Are there any differences in cellular sources of HMGB-1 or susceptible cells to HMGB-1 among the diseases.

Minor comments.

1. Page 1, line 33. Seventh leading cause of death in 2016 (WHO). Please update the statistical data.

2. Page 1, line 37. Ref. 4-9 include 2001, 2003, 2004 publications. “In the last few years” is inappropriate.

3. Page 3, line 116, Page 5, line215, table titles. Please correct “HMBG1”, and check all “HMGB1”.

4. Page 3, line 142. “HMGB-1” would be better than “HMGB-1 activity”.

5. age 4, line 148. In the chapter, author cited a review article Ref.54 as an original article.  

Author Response

Major comments

Authors describe that inflammatory stimuli enhance HMGB-1 expression or release in diabetes. However, inflammation is common pathophysiology in atherosclerosis and its complications. Please describe possible mechanisms of diabetes specific HMGB-1 expression and/or cell damage/death in macrovascular complications.

Authors’ response

We thank the Reviewer for this constructive comment. Additional paragraphs about possible mechanisms involved have been included in the new version of the manuscript (see page 3, lines 127-130 and page 4, lines 170-175).

Authors simply summarize effects of HMGB-1 on diabetic complications in Figure 1. Macrovascular, microvascular, and diabetic cardiomyopathy show distinct pathology. Are there any differences in cellular sources of HMGB-1 or susceptible cells to HMGB-1 among the diseases.

Authors’ response

We thank the Reviewer for this careful comment. We agree with the Reviewer that the underlying mechanisms are several and distinct. However, the purpose of the figure was to summarize the most important mechanisms, without aiming to list them in detail. We believe that, as it stands, the figure can be useful to most readers. Furthermore, in order to better clarify the various pathways, we have included the tables.

Minor comments.

Page 1, line 33. Seventh leading cause of death in 2016 (WHO). Please update the statistical data. Page 1, line 37. Ref. 4-9 include 2001, 2003, 2004 publications. “In the last few years” is inappropriate. Page 3, line 116, Page 5, line215, table titles. Please correct “HMBG1”, and check all “HMGB1”. Page 3, line 142. “HMGB-1” would be better than “HMGB-1 activity”. age 4, line 148. In the chapter, author cited a review article Ref.54 as an original article. 

Authors’ response

We thank the Reviewer for these helpful comments. Sentences and mistakes have been revised as suggested.